# Meeting Unmet Needs for Stroke Rehabilitation in Rural Public Health: Explorative Economic Evaluation of Upper Limb Robotics-Based Technologies through a Capabilities Lens

Natasha Brusco [1,*], Andrea Voogt [2], Melissa Nott [3], Libby Callaway [1,4], Mae Mansoubi [5] and Natasha Layton [1,*]

1 Rehabilitation, Ageing and Independent Living (RAIL) Research Centre, Monash University, Melbourne 3199, Australia
2 Department of Allied Health & Community Services, Northeast Health Wangaratta, Wangaratta 3747, Australia
3 Three Rivers Department of Rural Health, Charles Sturt University, Wagga Wagga 2640, Australia
4 Occupational Therapy Department, Monash University, Melbourne 3199, Australia
5 Digital Health Innovation and Public Health, INTERSECT, Medical School, University of Exeter, Exeter EX1 2LU, UK
* Correspondence: natasha.brusco@monash.edu (N.B.); natasha.layton@monash.edu (N.L.)

**Abstract:** Rehabilitation technologies are rapidly evolving, presenting promising interventions for people with neurological impairments. Access to technology, however, is greater in metropolitan than rural areas. Applying a capabilities approach to this access issue foregrounds healthcare recipients' rights and personhood within the discourse on resource allocation. Within this context, this study aimed to investigate the economic viability of robotics-based therapy (RBT) in rural Victoria, Australia. A regional health network developed a model of care to provide equitable access to RBT following stroke. This explorative economic evaluation examined both the clinical and economic impact of RBT program implementation across six program iterations compared to 1:1 out-patient rehabilitation. While clinical outcomes were equivalent, the per patient RBT cost ranged from AUD 2681 (Program 1) to AUD 1957 (Program 6), while the per patient cost of usual care 1:1 out-patient rehabilitation, was AUD 2584. Excluding Program 1, the health service cost of usual care 1:1 out-patient rehabilitation was consistently higher, indicating that an established RBT program may be cost-effective, specifically providing less cost for the same effect. This research demonstrates the economic feasibility of delivering RBT in a regional public health stroke service. More broadly, it provided a reduction in the capability gap between rural and metropolitan stroke survivors by tackling an access disadvantage.

**Keywords:** stroke; robotics; assistive technology; capabilities approach; economic evaluation



## 1. Introduction

Rehabilitation health care is a merit good, and one to which citizens have rights enshrined in international and national conventions and statutes [1,2]. In reality, however, rehabilitation access and provision are influenced by many factors. These factors include historical values, for example focusing expenditure on health priorities of the time, such as the eradication initiatives to address tuberculosis or polio epidemics [3]. Different priorities have been demonstrated to apply to different types of impairments, and the current focus on the COVID-19 pandemic is yet another variable [4]. Economies of scale represent another influencing factor, such as locating high-cost tertiary services in high-density metropolitan areas where they can be accessed by and serve more of the population [5,6].

The increase in both the incidence, and societal burden of, stroke-related disability globally means that government investment in multidisciplinary stroke services is significant [7]. International guidance on stroke rehabilitation outlines the importance of early intervention to reduce disability and enable a return to meaningful participation in society [7]. Assistive technology products and services are frequently used interventions

in stroke rehabilitation [8]. Technology interventions that are specific to upper limb rehabilitation after stroke demonstrate clinical benefits; however, translation into clinical practice has been found to be poor and dependent on multiple systemic factors. These factors include staff training and capabilities to implement novel interventions and access to technologies [9].

This access issue highlights that, just as certain populations gain from health care priority setting, others lose out. Substantial evidence points to the poor healthcare provision for very remote populations, lower socioeconomic groups, First Nations people, less visible or well-understood impairments, and for people whose health care needs are intersectional in nature [8–10]. Neoliberal macro health policy does a poor job of rationing healthcare [11], and attempts to bring an equitable lens to public need and provision have long been made, particularly in rural health [12].

Traditionally, patients undergoing rehabilitation following a stroke receive care from a multidisciplinary team (MDT) which may include physicians, physiotherapists, occupational therapists, speech therapists, neuropsychologists and nurses. This multidisciplinary team collaborates closely to offer integrated, comprehensive rehabilitation [13]. People with physical impairments from stroke usually undergo conventional physiotherapy or occupational therapy, which involves the person undertaking repetitive, high-intensity, task-specific exercises that enable them to regain their motor and functional abilities [14].

More recently, robotic devices have been introduced to rehabilitation to assist therapists by providing intensive, consistent and repetitive cycles over long periods to train the impaired limbs of patients. In terms of upper limb outcomes, systematic reviews have found that robot-assisted arm training improved arm motor movement and activities of daily living scores [15]. Overall, these reviews showed that at minimum, robotic devices offered equivalent treatment outcomes to conventional therapy and, in some patient groups, may be a more effective intervention choice [16]. However, the availability and use of robotics-based therapy (RBT) is impacted by various factors. These include the cost of and access to such technologies [10], staff training to ensure the safe and effective use of devices, and sustainability within service delivery models. A recent systematic review examined the cost of robotic rehabilitation following a stroke [15]. This review included five papers with a total of 213 patients. Based on these limited data, the review concluded that compared to usual care, robotics therapy may have cost advantages, particularly for patients with severe disability.

While robotic devices enable a high-intensity training regime that can be equally effective as conventional therapy, the robotic training equipment can cost up to several hundred to thousands of dollars per device, which is a significant capital outlay for health services. The decision to introduce robotic devices into clinical settings and offer robotic stroke rehabilitation to patients, therefore, requires an important cost consideration for health care providers. Despite its cost, it is surmised that robotic devices may increase the work efficiency of therapists. In turn, this offers the potential to treat more patients per therapist, or alternatively reduce the therapist hours required to provide therapy for the same patient numbers. Theoretically, this could lead to an overall reduction in the cost of treatment per patient or enable more patients to access therapy for the same cost.

The capabilities approach (CA) of Sen [17,18], developed further by Nussbaum [19,20] and explored by others [21–23], emerged from human development and economic roots. CA explicitly considers what people need and how these needs are resourced, using the language of capability (for example, freedoms, capabilities, and functioning). It also considers distributive justice and the achievement of rights, and can inform public policy [23]. CA offers a rights-based and person-centered perspective of the personal freedoms and capability gaps of stroke survivors based upon their rehabilitation access (including access to emerging technologies) and the impact of rurality [10,24]. Given these factors, within the context of CA, this study aimed to investigate the economic viability of robotics-based therapy used in upper limb stroke rehabilitation in rural Victoria, Australia.

## 2. Materials and Methods

The evaluation design included four phases (Figure 1).

### 2.1. The Setting

Sitting in wheat and pastoral country in the far northeast of Victoria, Australia, Northeast Health Wangaratta (NHW) provides health care to more than 90,000 people across North East Victoria. Previously, people residing in the northeast of Victoria requiring upper limb rehabilitation were offered conventional therapy at NHW or, to access RBT, needed to travel to the state's capital city of Melbourne, some 3.5 hours via car or train.

### 2.2. The Assistive Technology Implemented

In 2018, in an act of leadership in regional health care, NHW led an initiative through community fundraising and philanthropic grants to implement a safe, evidence-based model of care for RBT in neurological populations within its regional health service. This entailed the purchase of RBT equipment and related assessment kits and the establishment of a model of care for RBT, which included standardised protocols; valid and reliable outcome measures; competency-based training of staff; and ascertaining dosage recommendations (frequency and intensity of therapy).

### 2.3. The Economic Evaluation Approach

The explorative economic evaluation examined the clinical and economic impacts of implementing the RBT program. The clinical implications included the impact of the RBT program on both function and quality of life. The economic impact included the cost of participating in the RBT program from a limited societal perspective, inclusive of the regional public health service.

### 2.4. Ethical Considerations

The health service Human Research Ethics Committee deemed that ethics approval was not required to evaluate the RBT program, as this was a quality improvement project aiming to embed RBT into usual care; however, a clause was included in the patient behavioural contract (signed prior to commencing RBT) stating that the program was being evaluated and some de-identified data from the program would be used for the evaluation. This economic evaluation was prepared with reference to the Consolidated Health Economic Evaluation Reporting Standards (CHEERS) Checklist [25]. All costs are presented in AUD 2018/19.

### 2.5. Phases of the Analysis

2.5.1. Phase 1: Clinical Impact of the Robotics Program Compared to Usual Care

Intervention

The prospective upper limb robotics cohort included patients who participated in one or more of the six RBT program cycles during the data collection phase. Each RBT program cycle ran for six weeks, with three group sessions provided each week. While the RBT sessions ran for two hours, patients participated in 1.5 hours of the session, allowing the patients to be staggered and receive more individual attention from the program therapists. The RBT program was provided by a combination of qualified allied health professionals (physiotherapists and occupational therapists) and allied health assistants.

Data collection commenced in January 2018 with the commencement of the first of the six RBT programs and concluded in September 2018 with the conclusion of the last of the six RBT programs. Baseline and final assessments were initially conducted as a standalone session, in addition to the six-week program, but as therapy processes became more familiar and efficient, this transitioned to being completed within the program's usual sessions. Data were collected via a purpose-built data collection form and then entered into a purpose-built database.

**1) Clinical impact of the robotics program compared to usual care**

Robotics cohort: n=20. Prospective data collection across 6 robotics programs for incremental gain in function and quality of life

Usual care cohort n=10. Retrospective data collection via medical record audit for incremental gain in function and quality of life

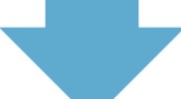

**2) Economic impact of the robotics program compared to usual care from a regional public health service perspective**

Robotics cohort: n=20. Prospective data collection across 6 robotics programs to represent the incremental cost of participation per patient (27 hours of intervention; 6 weeks x 3 sessions per week x 1.5 hour sessions)

Usual care: Modelled economic data to represent the incremental cost per patient receiving 27 hours of 1:1 out-patient upper limb rehabilitation intervention

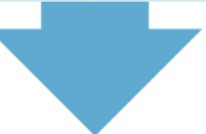

**3) Costs and opportunity costs associated with participation in the robotics program from the patient and carer perspective**

Robotics cohort: n=20. Prospective data collection across 6 robotics programs for the costs associated with participation in the robotics program, including travel costs, productivity opportunity costs for the carer and out-of-pocket fees for the robotics program

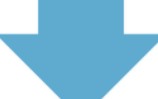

**4) Description of the cost effectiveness of the robotics program compared to usual care**

**Figure 1.** The four phases of the economic evaluation.

The usual care retrospective cohort was identified by the clinicians within the project team and included ten patients who had previously accessed the NHW Community Care Centre for rehabilitation of the upper limb post-stroke. For each patient, the medical record was accessed, and data were extracted using the RBT program data collection form. Sections of the data collection form that were not relevant to the usual care retrospective cohort were left blank. As there were minimal rehabilitation documentation standards in place at the time the retrospective cohort participated in the usual care (pre-robotic) rehabilitation program, it was expected that some data would be missing from medical records.

Sample size

As this was an explorative economic evaluation, the sample sizes for the robotics cohort and the usual care cohort were based on a sample of convenience. All patients involved in the first six RBT programs constituted the robotics cohort. A pre-determined retrospective sample of 10 patients constituted the usual care cohort.

Outcome measures

Demographic and injury-related information was extracted from the patient's medical record and entered into the project-specific database. Outcome measures included the Functional Independence Measure (FIM) as a measure of function [26] and the EuroQOL 5 dimensions three levels (EQ5D3L) as a measure of the quality of life [27] with answers converted into a health-related quality of life utility score using Australian weights [28]. The FIM and the EQ5D3L were collected at the commencement and conclusion of the RBT program for the robotics cohort and (when available) were extracted from the medical file recordings at the commencement and conclusion of out-patient rehabilitation for the usual care cohort.

2.5.2. Phase 2: Economic Impact of the Robotics Program Compared to Usual Care

Baseline health service and out-patient therapy utilisation (excluding the robotics program)

To establish the baseline health service and therapy utilisation for the two cohorts (excluding the RBT program), these data were collected for the robotics cohort and the usual care cohort from the onset of stroke until the conclusion of outpatient therapy. The purpose of collecting the baseline health service and therapy utilisation data was to understand the previous cost of care for the patient cohorts—that is, prior to commencing RBT. The costs were broken down into acute hospital costs, rehabilitation hospital costs and out-patient rehabilitation costs. The acute hospital costs and the rehabilitation hospital costs were modelled from the Independent Hospital Pricing Authority data on National hospital costs from the Round 20 Financial Year 2015/16 report [29] with the cost per day multiplied by the consumer price index (CPI) [30] to represent AUD 2018/19 (Table 1). Out-patient rehabilitation costs were modelled from the 2018 Transport Accident Commission rate for a 45- to 60-minute out-patient occupational therapy session also representing AUD 2018/19 (Table 2).

**Table 1.** Key classifications for the health service salary calculator.

|  | Classification | Code | Base Hourly | Total Hourly + 20.5% On-Costs |
|---|---|---|---|---|
| Grade 1 Clinician | Grade 1 Year 4 | VA5 | AUD 32.78 | AUD 39.52 |
| Grade 2 Clinician | Grade 2 Year 2 | VB2 | AUD 39.30 | AUD 47.39 |
| Grade 3 Clinician | Grade 3 Year 2 | VB8 | AUD 44.89 | AUD 54.13 |
| Allied Health Assistant | Grade 3 | TC2 | AUD 28.13 | AUD 33.93 |

**Table 2.** Cost data—units of measure for health care utilisation.

| Element | Unit Description/Data Source | Unit | Cost Per Unit |
|---|---|---|---|
| Pre-robotics health service costs, AUD 2018/19 |  |  |  |
| Acute admission | Cost of an acute hospital admission per diem [29] | 1 day | AUD 2104.76 |
| Rehabilitation admission | Cost of a rehabilitation hospital admission per diem [29] | 1 day | AUD 1124.36 |
| Out-patient sessions | Cost of an out-patient rehabilitation session with an allied health professional for 45 to 60 minutes duration [31] | 1 hour | AUD 94.96 |

Where available, health service utilisation was based on health administration data for both cohorts, and this was substantiated by patient recall for the prospective robotics cohort. Out-patient therapy utilisation was reported in hours, with a standard therapy session assumed to be of one hour in duration. The utilisation of health service acute and rehabilitation inpatient services was reported in days.

The cost analysis of the robotics program

The cost analysis of the RBT program was based on the cost of the RBT therapy staff, specifically the physiotherapist, occupational therapist and allied health assistant, as well

as the capital costs of the RBT equipment, maintenance and robotic consumables, the provision and receipt of staff education and the capital cost of space.

Data to calculate the staff costs for the delivery of each of the six RBT programs were collected using the purpose-built data collection form and database. It included the classification or grading of each staff member and the total minutes that staff were in attendance. The total minutes were used to calculate the costs using the health service salary calculator. The health service salary calculator provides the hourly rate of each staff member, based on their classification, and this is aligned with the relevant Enterprise Bargaining Agreement. The salary calculator included all relevant on-costs, and this equalled 20.5% of the base salary with the key salary classifications presented in Table 1.

The set-up costs for the RBT equipment were based on the 2017/18 net worth of the RBT equipment inflated by CPI for a 2018/19 NPV, with an additional 25% added for the cost of maintenance and consumables associated with the equipment (Table 3). The specific types of robotics equipment are detailed in Table 3. It was expected that the equipment would service the program for four years. Therefore, the cost of the RBT equipment was deducted across four years with a pro-rata cost allocation for each six-week RBT program. Capital costs for space were included in the RBT program, and this was based on a daily rate for each RBT session. This was modelled from a previous study with the AUD 2010/11 cost per day in the previous study multiplied by CPI to represent AUD 2018/19 [32]. Table 3 reports the data source, unit description and unit cost for the RBT equipment, maintenance and consumables, the provision of staff education, the receipt of staff education and the cost of space.

**Table 3.** Cost data—robotics program set up costs.

| Set Up Cost Element, AUD 2018/19 | Unit Description/Data Source | Unit | Cost Per Unit (s) |
|---|---|---|---|
| Capital costs of equipment, AUD<br>Amadeo<br>Tymo<br>Pablo<br>Able X mouse and handbar<br>Saebo Re-joyce<br>Kidney shaped tables<br>High low tables<br>Laptops | Health service equipment purchase orders from 2017/18 with CPI applied for 2018/19 net present value [3] | 1 robotics device<br>1 robotics device<br>1 robotics device<br>1 robotics device<br>1 robotics device<br>2 tables<br>2 tables<br>3 laptops | AUD 85,764<br>AUD 10,618<br>AUD 12,252<br>AUD 2042<br>AUD 20,341<br>AUD 2042<br>AUD 2042<br>AUD 3063 |
| Maintenance and consumables associated with the equipment | 25% of the capital cost of equipment [33] | 25% of capital costs | AUD 35,645.91 |
| Provision of staff education, AUD | Health service salary calculator based on Grade 3 Year 4 wage classification for an OT/PT<br><br>1 hour = AUD 54.61<br>AUD 45.29 × 1.2057 (20.57% on-costs) | 8 hours preparation<br>2-hour workshop<br>2 × 1-hour group education<br>15 × 1-hour individual education<br>TOTAL 27 hours | AUD 1474.47 |
| Receipt of staff education, AUD | Health service salary calculator based on an average Grade 2 Year 1 wage classification for an OT/PT<br><br>1 hour = AUD 45.31<br>AUD 37.57 × 1.2057 (20.57% on-costs) | 2-hour workshop (15 attendees)<br>2 × 1-hour group education (10 attendees)<br>15 × 1-hour individual education<br>TOTAL 65 hours | AUD 2945.15 |
| TOTAL SET UP COSTS | | | AUD 178,229.53 |
| Cost of space | Each robotics program requires 18 half days of space (AUD 8.49 × 18 days × 1/2 day) for a total space cost of AUD 76.41 per program [33] | A 1/2-day session | AUD 4.25 |

The cost of usual care

Staffing costs for the usual care were based on the provision of 27 hours of 1:1 out-patient rehabilitation with an occupational therapist or physiotherapist (Grade 2 Year 2). This was to ensure therapy participation equity with the RBT program, which involved the provision of 27 hours of rehabilitation intervention (six-week program x three sessions per week × 1.5-hour sessions). The usual care costs included a salaries and wages (S&W) base-

rate of AUD 44.89 plus 20.5% on-costs for a total S&W of AUD 54.13 per hour (based on the health service salary calculator; see Table 1). In addition to S&Ws, another 75% loading was added to account for all non-S&W costs. This loading included clinical supervision, education, capital cost for space, and consumables utilised during therapy. This took the total hourly rate for usual care to AUD 94.96, which is consistent with the 2018 Transport Accident Commission rate for a 45- to 60-minute out-patient occupational therapy session. As such, participation in 27 hours of 1:1 out-patient rehabilitation incurred a total cost of AUD 2563.92 per patient from the regional public health service perspective.

2.5.3. Phase 3: Costs and Opportunity Costs from the Patient and Carer Perspective

In addition to the regional public health service perspective, data were collected and reported in relation to the costs and opportunity costs associated with participation in the RBT program that pertain to the patient and carer.

Travel costs for the patient

For the prospective robotics cohort, the travel costs for attending the program were calculated based on the distance between home and the rehabilitation Community Care Centre. The distance was doubled to represent a return trip, and the total distance in kilometers was multiplied by 68 cents to represent the cost of travel [3].

Productivity opportunity costs for the carer

Due to the nature of the neurological injury and ongoing restrictions on independence (particularly with respect to driving), all program participants were accompanied by a familial carer. Consideration of lost productivity is a critical and often unseen cost. With respect to carer productivity, the potential opportunity cost for the carer is the loss of a day of paid or unpaid work. The average earnings of an Australian per week are AUD 1586, or AUD 317 per day [3]. To attend all 18 sessions of the six-week program, there is a potential productivity loss of AUD 5710. Productivity opportunity costs for the patient were not included, as clinicians reported that the patient cohort involved in the RBT program had minimal or no employment (paid or unpaid) at the time of engaging in the program.

Robotics program out-of-pocket fees

Participation in the RBT program incurred a nominal fee of AUD 20 per week. This equated to a total of AUD 120 per patient over the duration of the six-week program. This fee was not enforced on all participants, with exceptions made for those patients already participating in the health service Early Supported Discharge program (this program also had a nominal fee), those deemed to have financial hardship, or other reasons.

2.5.4. Phase 4: Description of the Cost Effectiveness of the Robotics Program Compared to Usual Care

It was planned that the dominance of cost and effect would be described in the cost-effectiveness analysis, comparing RBT to usual care.

Assumptions

Care needs to be taken with regard to the interpretation and generalisability of these results, as there are several assumptions and limitations that underpin this economic evaluation. The assumptions include:

- That a patient receiving usual care 1:1 out-patient rehabilitation therapy would be able to access an intensity equivalent to the RBT program.
- That a patient receiving usual care 1:1 out-patient rehabilitation therapy would be managed by a Grade 2 clinician (or above).
- That an Allied Health Assistant completes the set up and pack up for the RBT program and the usual care 1:1 out-patient rehabilitation therapy, and that this has been included as a part of the program on-costs.
- That administrative staff complete all administration duties for the RBT program and the usual care 1:1 out-patient rehabilitation therapy, and that this has been included as a part of the program on-costs.

- That any costs associated with a home exercise program for both the RBT program and the usual care 1:1 out-patient rehabilitation therapy have been included as a part of the program on-costs.

Sample size, primary and secondary analysis

The primary analysis was based on the individual patients who completed the RBT program for the first time, with secondary analysis reporting on all admissions to the RBT program as well as reporting on the second and third admissions to the RBT program (as patients were able to repeat the program if clinically appropriate). In the case of missing data, participants were excluded case-wise. Changes for the FIM and the EQ5D3L were measured from the commencement to conclusion of outpatient therapy for the usual care cohort and from the commencement to conclusion of the RBT program from the robotics cohort, and these were analysed using independent *t*-tests. Cost data were analysed between groups using independent *t*-tests to examine between-group differences and test for significance ($p < 0.05$). If an outcome measure achieved or trended towards a significant difference, a cost-effectiveness and/or cost-utility analysis was planned using an incremental cost effectiveness ratio (ICER) derived from the mean difference (MD) in the outcome and the MD in the cost. Where an ICER was not possible, a qualitative description of cost-effectiveness was provided. All cost data were inflated to AUD 2018/19. A 5% level of significance was applied with 95% confidence intervals (CIs) to represent variance, unless otherwise stated.

## 3. Results

### 3.1. Clinical Impact of the Robotics Program Compared to Usual Care

Across the six RBT program cycles completed during the data collection period, there were 20 episodes of patient participation by 14 individual patients. In addition to participating in an initial RBT program, five patients participated a second time, and one patient participated a second and a third time. The 20 episodes represent the total prospective RBT program cohort. The retrospective usual care cohort included ten eligible participants with variable outpatient therapy start dates that ranged from December 2015 to July 2018.

Based on the 14 individual patients who participated in the RBT program for the first time, the robotics cohort and the usual care cohort may have been different at baseline. There was a trend towards the robotics cohort having a greater distance from home to the Community Care Centre (MD −22.2 km; $p = 0.10$) and having more time between the onset of the stroke and the baseline assessment (MD −0.45 years or −5.4 months; $p = 0.08$) (Table 4). The years from stroke onset to baseline assessment may be clinically meaningful, as the recovery rate may reduce over time. The significant difference between groups for the measure of the dominant hand being affected by the stroke may not be meaningful due to n = 8/10 missing values for the baseline group (Table 5).

**Table 4.** Patient outcomes.

| Outcome | Usual Care Cohort (n = 10) | | | Robotics Cohort All Participation (n = 20) | | | Robotics Cohort First Program Participation (n = 14) | | | Robotics Cohort Second or Third Program Participation (n = 6) | | | Difference between Groups for Change Score (First Time Robotics Minus UC) |
|---|---|---|---|---|---|---|---|---|---|---|---|---|---|
| | Baseline | Post Intervention | Baseline to Post Change Score | Baseline | Post Intervention | Baseline to Post Change Score | Baseline | Post Intervention | Baseline to Post Change Score | Baseline | Post Intervention | Baseline to Post Change Score | |
| FIM total, score (SD) | n = 4 111.8 (15.6) | n = 2 125.0 (1.4) | n = 2 2.5 (3.5) | n = 18 100.9 (13.4) | n = 15 102.5 (13.9) | n = 15 3.7 (5.1) *Usual care to ALL robotics* $p = 0.73$ | n = 12 99.7 (13.9) | n = 11 99.6 (13.4) | n = 11 2.1 (4.7) *Usual care to FIRST robotics* $p = 0.90$ | n = 4 103.5 (13.2) | n = 4 110.50 (13.6) | n = 4 8.0 (3.6) *Usual care to 2nd/3rd Robotics* $p = 0.21$ | 0.4 (95%CI −14.1 to 15.0) $p = 0.90$ |
| Euro QOL 5D3L, VAS (SD) | NA | NA | NA | n = 19 70.26 (15.4) | n = 18 69.11 (18.6) | n = 18 −1.15 95%CI (−10.29 to 12.60), $p = 0.839$ | n = 13 68.4 (15.5) | n = 13 71.8 (17.3) | n = 12 1.25 95%CI (−4.0 To 6.5) $p = 0.61$ | n = 6 74.3 (15.8) | n = 5 62.2 (22.0) | n = 5 −9.0 95%CI (−40.2 To 22.2), $p = 0.47$ | NA |
| Euro QOL 5D3L, utility index (SD) | NA | NA | NA | n = 20 0.63 (0.1) | n = 18 0.63 (0.1) | n = 18 0.001 95%CI (−0.077 to 0.737), $p = 0.970$ | n = 14 0.63 (0.09) | n = 13 0.62 (0.9) | n = 13 −0.001 95%CI (−0.07 to 0.07) $p = 0.99$ | n = 6 0.63 (0.17) | n = 5 0.66 (0.12) | n = 5 0.010 95%CI (−0.22 to 0.24) $p = 0.91$ | NA |

**Table 5.** Patient demographics.

| Characteristic | Usual Care Cohort (n = 10) | Robotics Cohort (n = 20) | | | Mean Difference and Significance (Usual Care to First Time in the Robotics Program) |
|---|---|---|---|---|---|
| | | All Participation (n = 20) | First Program Participation (n = 14) | Second or Third Program Participation (n = 6) | |
| Age, years (SD) | 71.0 (10.7) | 66.3 (7.3) | 66.3 (7.9) | 66.4 (6.6) | 4.8 years $p = 0.23$ |
| Gender, n (%) Female | 3 (30%) | 12 (60%) | 7 (50%) | 5 (83%) | $p = 0.32$ |
| Years from onset to baseline assessment, years (SD) | 0.21 (0.33) | 0.84 (1.29) | 0.67 (0.79) | 1.2 (2.1) | −0.45 years $p = 0.08$ |
| Distance from home to community care centre, KMs (SD) | 27.6 (27.7) | 47.7 (30.1) | 49.9 (35.6) | 42.6 (10.4) | −22.2 km $p = 0.10$ |
| Transport, n (%) Carer driven car Self–driven car Walk Other | 8 (80%) 0 (0%) 1 (10%) 1 (10%) | 16 (80%) 1 (5%) 0 (0%) 3 (15%) | 11 (79%) 0 (0%) 0 (0%) 3 (21%) | 5 (83%) 1 (17%) 0 (0%) 0 (0%) | $p = 0.53$ |
| Diagnosis | 10 (100%) | 20 (100%) | 14 (100%) | 6 (100%) | $p = 1.00$ |
| Dominant hand affected, n (%) yes | 1 (10%) (Unknown n = 8) | 9 (45%) | 6 (43%) | 3 (50%) | $p = 0.00$ * |
| Pre-existing UL condition, n (%) yes | 1 (10%) | 4 (20%) | 3 (21%) | 1 (17%) | $p = 0.26$ |

* Statistically significant $p < 0.05$.

As each iteration of the program passed, there were minor modifications to the program. This includes a change in the staffing profile (from more senior to less senior staff present), an overall reduction in staff present, as well as more patients attending per program. (Figures 2 and 3).

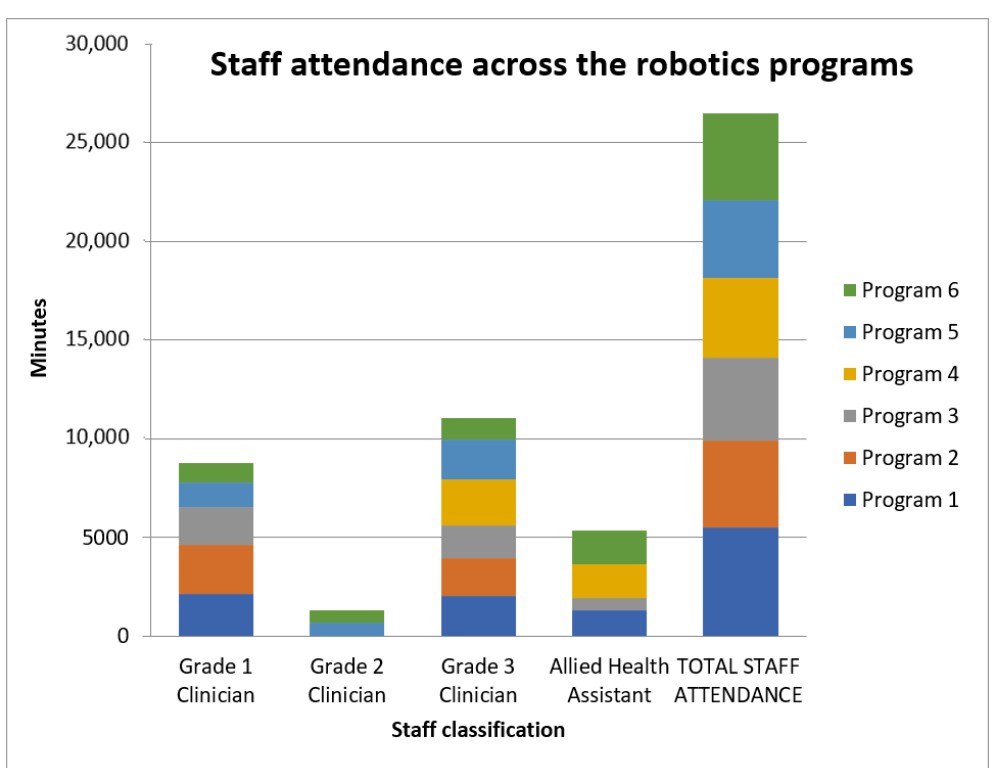

**Figure 2.** Staff attendance across the RBT program.

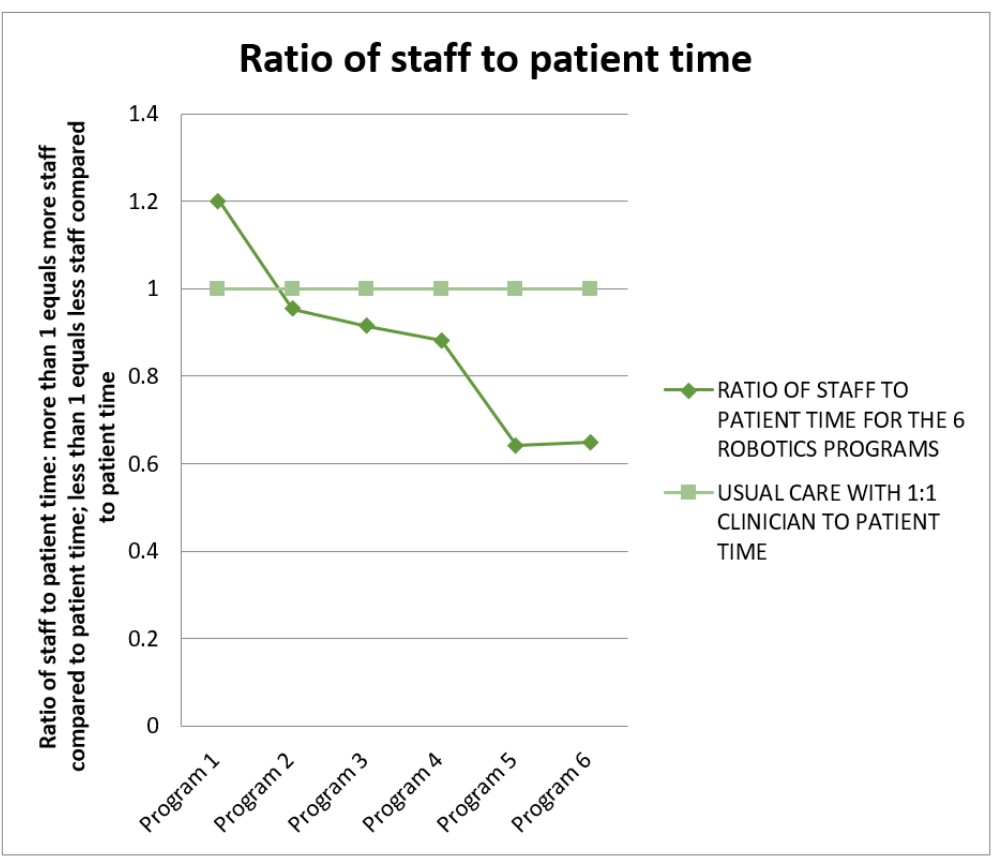

**Figure 3.** Ratio of staff to patient time.

3.1.1. Outcome Measures

The RBT program is presented as all programs (n = 20), as first program (n = 14) and as second or third program (n = 6) to capture the patients in this cohort who entered the program on more than one occasion. Those in the first program (n = 14) are considered in the primary analysis.

The FIM demonstrated observed improvements in the functional score for both cohorts from baseline to post-intervention, yet there were no within-group, between-group or sub-group scores that achieved or trended towards statistical significance. The usual care group had 70% to 80% missing data for this outcome.

The EuroQOL5D3L demonstrated no differences from baseline to post-intervention for the robotics cohort in the all programs (n = 18), first program (n = 12) and second or third program (n = 5) groups. The usual care group had 100% missing data for this and any QOL outcome.

3.1.2. Economic Impact of the Robotics Program

Health service and outpatient therapy utilisation were collected for both the prospective robotics cohort and the usual care participants, excluding the cost of the RBT program, to compare utilisation. While there was an observed reduction in acute admissions and outpatient session utilisation and cost for the robotics cohort, there was a significant reduction in rehabilitation admissions and cost for the usual care cohort (Tables 6 and 7).

**Table 6.** Baseline health service and out-patient therapy utilisation.

| Element | Usual Care Cohort (n = 10) | Robotics Cohort (n = 14) | Mean Difference (95%CI) | Significance |
|---|---|---|---|---|
| Pre-robotics total (SD) Acute admission—days | 12.90 (14.19) | 7.36 (6.72) | 6.05 (−4.320 to 16.42) | $p = 0.27$ |
| Rehabilitation admission—days | 17.90 (23.39) | 50.00 (42.92) | −29.85 (−54.269 to −5.431) | $p = 0.03$ * |
| Out-patient sessions—hours | 28.50 (15.47) | 21.43 (21.79) | 6.75 (−7.17 to 20.67) | $p = 0.36$ |

* Statistically significant $p < 0.05$.

**Table 7.** Baseline health service and out-patient therapy cost.

| Element | Usual Care Cohort (n = 10) | Robotics Cohort (n = 14) | Mean Difference (95%CI) | Significance |
|---|---|---|---|---|
| Pre-robotics total cost, AUD (SD) | | | | |
| Acute admission | AUD 27,151 (29,872) | AUD 15,485 (14,146) | AUD 11,666 (−10,523 to 33,856) | $p = 0.27$ |
| Rehabilitation admission | AUD 20,126 (26,296) | AUD 56,218 (48,260) | −AUD 36,092 (−68,020 to −4164) | $p = 0.03$ * |
| Out-patient sessions | AUD 2706 (1469) | AUD 2035 (2069) | AUD 672 (−826 to 2169) | $p = 0.36$ |
| Total cost | AUD 49,984 (48,681) | AUD 73,378 (56,039) | −AUD 23,754 (−68,414 to 20,905) | $p = 0.28$ |

* Statistically significant $p < 0.05$.

### 3.1.3. Cost Analysis

The intended cost effectiveness analysis was not completed due to significant limitations with the available data, which presented a high risk of bias for internal validity. This included the small sample size, differences between groups at baseline, and between 70% and 100% missing data for key outcomes for the usual care group.

As such, the following section is presented as a cost analysis (this is a direct comparison of costs) of the RBT program (which offers each patient 27 hours of upper limb RBT therapy based on a six-week program, three sessions per week and 1.5 hours per session), compared to 27 hours of 1:1 out-patient rehabilitation (modelled data). This assumption for economic modelling was confirmed as an appropriate choice, as it was reported earlier in the results section that the usual care cohort had an average of 29 hours of 1:1 outpatient therapy in the existing out-patient services at NHW prior to the implementation of the RBT program.

### 3.1.4. Capital Costs for the Robotics Equipment and Education Costs for the Robotics Program

The set-up costs for the RBT equipment were based on the net cost of the RBT equipment, as well as the cost of the specialised tablets and laptops (Table 4). The total cost for the RBT equipment, including tablets and laptops, and the cost of maintenance and consumables associated with the equipment, was AUD 173,810. When depreciated over four years, this is an annual cost of AUD 43,452. The current study included six iterations of the RBT program. Each iteration of the program ran for six weeks, representing six months of RBT utilisation (noting the overlap between some of the six-week periods). Therefore, half of the annual cost (AUD 43,452/2 = AUD 21,726) can be distributed across the six programs for a cost of AUD 3621 per 6-week program.

The set-up costs for the education included the staff costs for the provision of the education as well as the staff costs for receipt of the education (Table 4). This represents that total education required for the six × 6-week programs over four blocks (24 weeks). The total cost of education was AUD 4419, and this can be distributed across the six × 6-week programs for a cost of AUD 737 per 6-week program. It is likely that the cost of education per program would reduce over time, as it is assumed that the preparation time for the education material and the number of new staff requiring education on an ongoing basis would be less than that required for the initial six × 6-week programs. The cost of space for the RBT program is AUD 8.49 per day. Each RBT program requires 18 half days (AUD 8.49 × 18 days × 1/2 day) for a total space cost of AUD 76.41 per program.

Total capital costs, education costs and space costs were a combined AUD 4434 per program, and with an average of 3.3 participants per program, this equates to AUD 1344 per participant per program.

### 3.1.5. Staff Costs for the Robotics Program and for Usual Care

Based on a 6-week program with three × 1.5-hour therapy sessions per week, the RBT program provides 27 hours of intervention. To provide 27 hours of out-patient rehabilitation via a traditional 1:1 clinician to patient usual care model (based on a Grade 2 clinician), the cost of salaries and wages would be AUD 1919 per patient (Table 8). While this would be less for a Grade 1 clinician (AUD 1601) and more for a Grade 3 clinician (AUD 2192), the costs savings for a junior member of staff would be off-set with the additional costs associated with greater clinical supervision, education and support.

**Table 8.** Staff costs for each of the 6 RBT programs and for usual care.

| | Program 1 | Program 2 | Program 3 | Program 4 | Program 5 | Program 6 | USUAL CARE |
|---|---|---|---|---|---|---|---|
| **Patient group time per program** | 2040 | 2040 | 2040 | 2040 | 2040 | 2160 | 2430 |
| **Number of sessions per program** | 17 | 17 | 17 | 17 | 17 | 18 | 18 |
| **Patients enrolled per program** | 3 | 3 | 3 | 3 | 4 | 5 | 1 |
| **Patient session attendance (sessions)** | 51 | 51 | 51 | 51 | 68 | 75 | 18 |
| **Patient session attendance (time)** | 4590 | 4590 | 4590 | 4590 | 6120 | 6750 | 2430 |
| **STAFF SESSION ATTENDANCE** | | | | | | | |
| **Grade 1** | 18 | 20.5 | 16.25 | 0 | 10 | 8.25 | 0 |
| **Grade 2** | 0 | 0 | 0 | 0 | 6 | 5 | 18 |
| **Grade 3** | 17 | 16 | 13.75 | 19.5 | 16.75 | 9 | 0 |
| **AHA** | 11 | 0 | 5 | 14.25 | 0 | 14.25 | 0 |
| **TOTAL STAFF SESSION ATTENDANCE** | 46 | 36.5 | 35 | 33.75 | 32.75 | 36.5 | 18 |
| **STAFF ATTENDANCE IN MINUTES** | | | | | | | |
| **Grade 1** | 2160 | 2460 | 1950 | 0 | 1200 | 990 | 0 |
| **Grade 2** | 0 | 0 | 0 | 0 | 720 | 600 | 2430 |
| **Grade 3** | 2040 | 1920 | 1650 | 2340 | 2010 | 1080 | 0 |
| **AHA** | 1320 | 0 | 600 | 1710 | 0 | 1710 | 0 |
| **TOTAL STAFF ATTENDANCE IN MINUTES** | 5520 | 4380 | 4200 | 4050 | 3930 | 4380 | 2430 |
| **RATIO OF STAFF TO PATIENT TIME** | 1.2 | 1.0 | 0.9 | 0.9 | 0.6 | 0.6 | 1.0 |
| **STAFF COST (S&W) FOR ATTENDANCE** | | | | | | | |
| **Grade 1** | AUD 1423 | AUD 1620 | AUD 1284 | – | AUD 790 | AUD 652 | – |
| **Grade 2** | – | – | – | – | AUD 569 | AUD 474 | AUD 1919 |
| **Grade 3** | AUD 1840 | AUD 1732 | AUD 1489 | AUD 2111 | AUD 1813 | AUD 974 | – |
| **AHA** | AUD 746 | – | AUD 339 | AUD 967 | – | AUD 967 | – |

**Table 8.** *Cont.*

|  | Program 1 | Program 2 | Program 3 | Program 4 | Program 5 | Program 6 | USUAL CARE |
|---|---|---|---|---|---|---|---|
| **TOTAL STAFF COST (S&W) FOR ATTENDANCE** | AUD 4010 | AUD 3352 | AUD 3112 | AUD 3078 | AUD 3172 | AUD 3067 | AUD 1919 |
| **TOTAL STAFF COST (S&W) PER PATIENT** | AUD 1337 | AUD 1117 | AUD 1037 | AUD 1026 | AUD 793 | AUD 613 | AUD 1919 |

The cost of salaries and wages for the RBT program ranged from AUD 1337 (Program 1) to AUD 613 (Program 6) per patient (Table 8). With each subsequent RBT program, the cost of salaries and wages per patient reduced (Figure 4). This demonstrates efficiency over time, with the need for less direct staff input into the program as well as the use of more junior classifications to staff the program, combined with an increase in the number of patients attending each program.

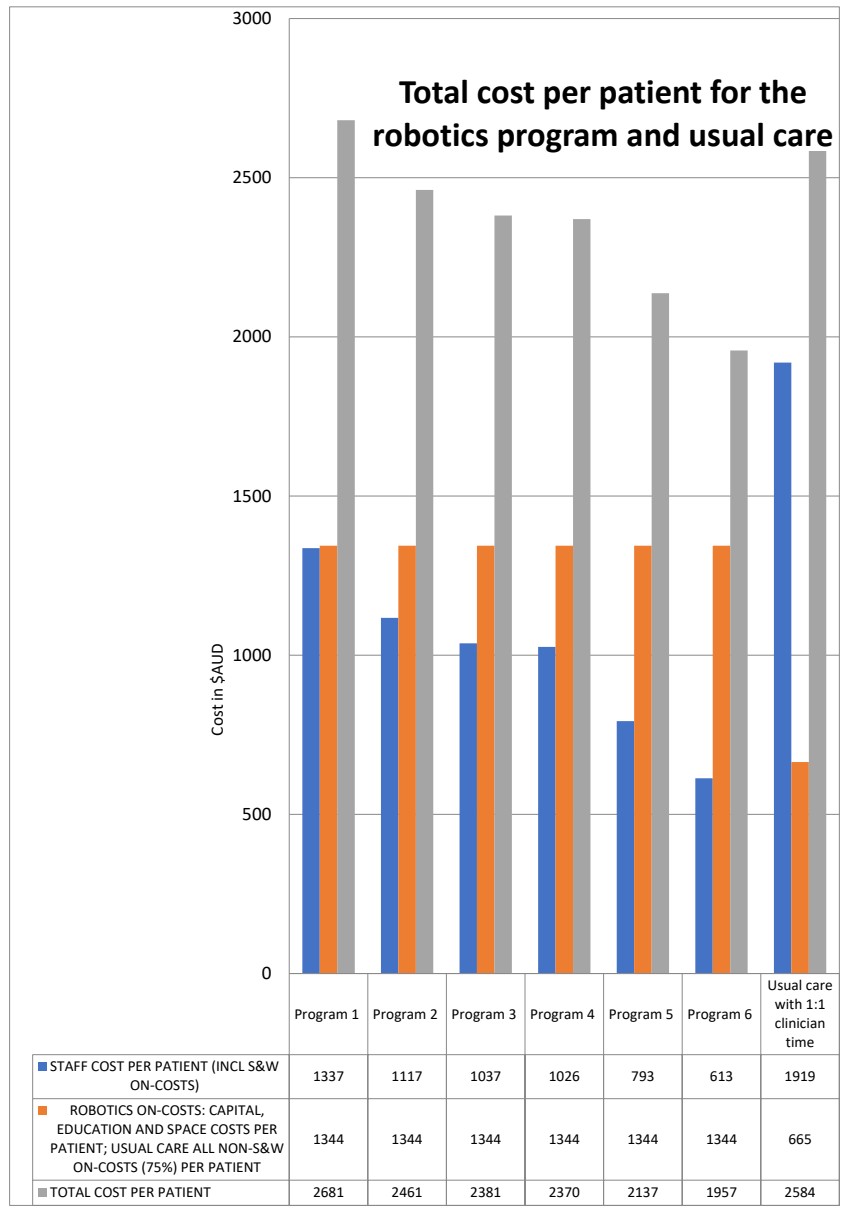

**Figure 4.** Total cost per patient.

The ratio of staff to patient time followed the same pattern as the cost of salaries and wages for the RBT program, with each subsequent RBT program reducing or maintaining the ratio of staff to patient time (range 1.2:1 to 0.6:1). This is compared to usual care out-patient rehabilitation which requires a 1:1 clinician to patient ratio (Figure 3). This is consistent with the staff cost per patient (Figure 5).

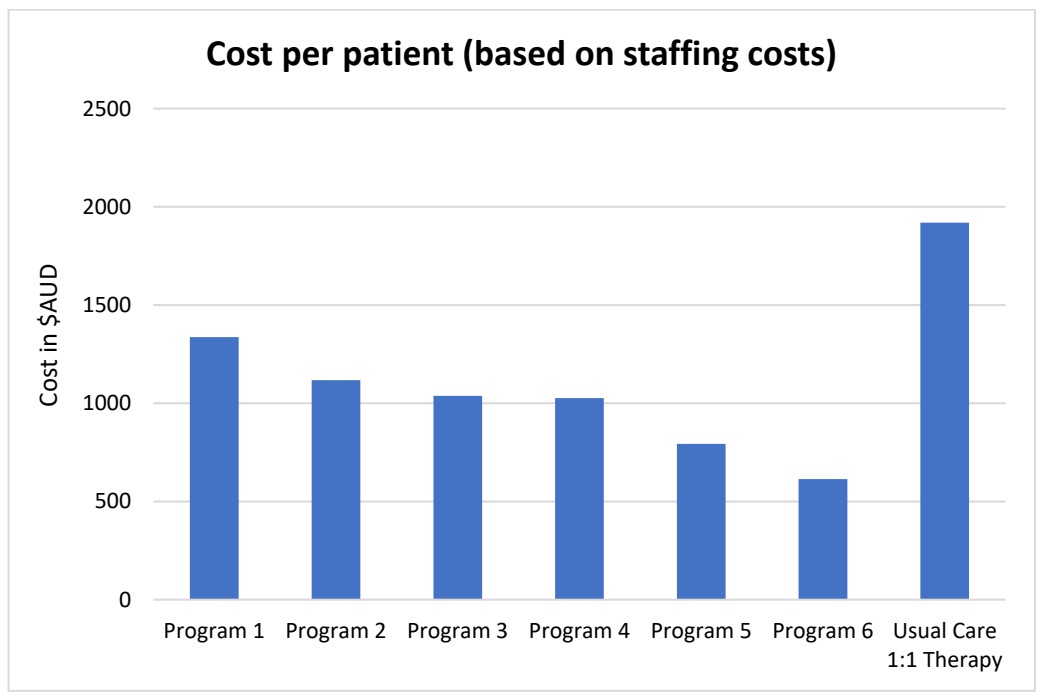

**Figure 5.** Staff cost per patient.

Per patient, the total cost of the RBT program ranged from AUD 2681 to AUD 1957; with each subsequent RBT program, the total cost per patient reduced (Figure 4). Per patient, the total cost of usual care 1:1 out-patient rehabilitation was AUD 2584 (Figure 4). Excluding the first of the RBT programs (Program 1), the cost of usual care 1:1 out-patient rehabilitation was consistently higher for the health service.

### *3.2. Costs and Opportunity Costs from the Patient and Carer Perspective*

#### 3.2.1. Travel Costs for the Patient

One participant completing the RBT program for the second time transitioned from transport where the carer drove the car (first time in program) to transport that was a self-driven car (second time in program). This was the only occasion when a participant drove themselves to the RBT program. For the prospective robotics cohort, the cost of travel for attending the RBT program by car was AUD 1221 (SD AUD 872). This is based on an average distance of 50 km from home to the health service, representing a 100 km return trip, and attending all 18 sessions of the 6-week program.

#### 3.2.2. Productivity Opportunity Costs for the Carer

The "patient mode of transport" that is reported in Table 5 reports that 80% of the time, the carer was required to drive the patient to and from the RBT sessions. In order to provide transport to the patient, the carer would need to forgo any paid or unpaid activity that would potentially contribute to their productivity. Based on the average Australian wage of AUD 317 per day, to attend all 18 sessions of the 6-week program, there is a potential productivity loss of AUD 5710 for the carer for each 6-week RBT program, and this potential loss of productivity is applicable to 80% of the carers supporting the participants of the RBT program.

### 3.2.3. Robotics Program Out-of-Pocket Fees

The RBT program has a nominal fee of AUD 20 per week. The application of this fee varied from week to week of the program and varied between each of the six iterations of the program (Table 9). On average, each program had 3.3 participants, and of these participants, an average of two participants paid the AUD 20 weekly fee each week. This provided each program with AUD 240 of revenue from patient out-of-pocket fees.

**Table 9.** Application of the robotics program weekly out-of-pocket fee.

|  | Week 1 | Week 2 | Week 3 | Week 4 | Week 5 | Week 6 |
|---|---|---|---|---|---|---|
| Robotics weekly fee applied | 14 | 9 | 13 | 15 | 15 | 5 |
| Fee excluded due to Early Supported Discharge program | 1 | 2 | 3 | 2 | 2 | 3 |
| Fee excluded due to financial hardship or other reason | 5 | 9 | 4 | 3 | 3 | 12 |

### 3.2.4. Description of the Cost Effectiveness of the Robotics Program Compared to Usual Care

As a cost-effectiveness analysis was not appropriate due to no significant or trended differences in the clinical effect related to the RBT program, the cost effectiveness of the program is qualitatively described. The initial iteration of the RBT program was more costly than the provision of an equivalent intensity of 1:1 out-patient rehabilitation therapy. This is likely due to the presence of senior staff and the high staffing ratio present during the first program while the staff were implementing the program for the first time. However, subsequent iterations of the RBT program were less costly than the provision of an equivalent intensity of 1:1 out-patient rehabilitation therapy (Figure 4). This indicates that an established RBT program may be cost-effective when compared to the provision of an equivalent intensity of 1:1 out-patient rehabilitation therapy, specifically requiring less cost for the same effect.

### 4. Discussion

Rehabilitation technologies are rapidly evolving and represent a promising intervention for individuals experiencing impairment from stroke or other health conditions. The uptake of technologies into the field, and provision to those who would benefit, is, however, subject to many factors. These include both the cost of the technology as well as access to it [9]. Social justice approaches, such as the capability approach, relocate problems—including rehabilitation access—away from the individual. This allows for the social model of disability and the notion of disablement by the environment, and the consideration of the policy or funding environment and its impact on health services [23,34]. Applying a CA lens in this exploratory economic analysis of RBT foregrounds the rights and personhood of healthcare recipients who have experienced stroke within the discourse of resource allocation, and challenges the notion that costly or innovative technologies such as RBT are out of reach [23].

A recent systematic review of robotics rehabilitation economic evaluations identified five studies of moderate methodological quality [15]. Interventions spanned the acute, subacute and chronic stages of rehabilitation following a stroke, and all included upper limb rehabilitation, with one including both upper and lower limb rehabilitation [15]. The results indicate that for the robotics intervention groups, while the cost was generally less (four of the five included studies), both groups achieved that same health benefit (four of the five included studies) [15]. This finding is consistent with the current study, where—after program establishment—there was a lesser cost per patient in the robotics group, whilst the same health benefit was achieved. It is noted that this explorative economic evaluation only includes upper limb robotics. A recent systematic review of lower limb (or gait) robotics,

combined with traditional physiotherapy and body weight support treadmill practice, was also found to be an efficient mode of therapy delivery after stroke [35].

This growing body of RBT evidence highlights the importance of access to contemporary rehabilitation, as well as the use of economic evaluation methods to inform government policy and investment in emerging technologies [1]. The current study adds new evidence by coupling a CA lens with an economic evaluation framework. This approach has allowed the investigation of ways to measure the gap between functioning, what people are able to do and be, and capabilities—that is, the opportunities stroke survivors may have or can choose from as part of their rehabilitation journey [23].

Access for stroke survivors to technology-mediated therapy has consistently enabled greater engagement in therapy, leading to positive clinical outcomes [36]. Using the capabilities approach, rehabilitation program evaluations may more closely consider the opportunities for human agency. In the current study, by using RBT, patients experienced greater access to meaningful rehabilitation activities that promoted agency and engagement. RBT may therefore be framed as a mediator of the capability gap for stroke survivors. Although exploratory in nature, this research therefore begins to inform a rights-based and person-centred perspective to address the potential capability gap for stroke survivors based upon their rehabilitation access (including the distributive justice offered by access to RBT) [10,24]. This research also provides an example of ways to evidence-inform policies for people with a disability, including those who have experienced stroke, via a CA framework [23].

There do, however, continue to be various challenges for both people with stroke and health care professionals working in stroke rehabilitation, and these are heightened by access issues in regional and rural areas [16,37,38]. Specific to health services, challenges include delivering services that are both efficient and effective, maintaining the currency of practice with the rapid emergence and diversification of rehabilitation technologies and design paradigms, the availability of specialists skills/knowledge/services, the access to staff training to advance staff capabilities, and managing the socio-political factors that influence technology supply and practice [34] (p.668). Importantly, the current study provides an economic evaluation framework which demonstrates that, as RBT became more efficient, the cost per patient reduced to below the cost per patient for usual care. This evidence may offer opportunities for health networks—and health professionals working in them—to enact systemic advocacy for investment in robotic technologies in order to achieve the best possible outcomes for people with upper limb impairment from stroke [34].

As with the literature that identified robotic rehabilitation training for patients as being less expensive than conventional training [15], the results of the current economic evaluation demonstrate the overall costs to be less expensive for the robotic program compared to the usual care cohort. It also showed that efficiency improvements exist as the number of programs delivered increase. In addition to the financial benefit experienced by the health service in this study, the introduction of RBT programs provided opportunities for allied health staff to engage in specialist training and education in order to build their own skills and capabilities in delivering novel technologies. This program also offered staff the opportunity to develop skills in data collection and contribute to quality assurance/program evaluation activities, responding to international calls for action towards strengthening rehabilitation in health systems [1].

To further enhance sustainability and improve financial viability, rural and regional health services may consider opportunities to increase the number of participants per program. The economic evaluation demonstrated that increasing the number of participants reduces the overall cost per participant compared with usual care. The cost of the later programs (Program 5 and 6) clearly demonstrated reduced staff costs per patient due to the change in skill mix and overall hours. The consistent application of the out-of-pocket patient co-payment would further improve the financial viability. Consideration might also be given to the appropriate price point for a financial contribution by participants. Alternative opportunities to increase the number of program participants and achieve greater program

access could extend to exploring portable robotic devices. This evaluation focused on a centre-based model in which patients travel to and from a community centre to engage in RBT; however, in a rapidly expanding field, portable devices for home-based use by stroke survivors are becoming increasingly available [39]. For patients living in rural and remote locations, home-based therapy can minimise traditional barriers to service access and can promote greater autonomy, enhance recovery and achieve greater outcomes [40]. Hybrid models of centre-based and home-based rehabilitation can offer enhanced sustainability via reduced staff contribution. In particular, utilising technologies such as telehealth to offer virtual care to the home setting may be clinically appropriate and cost effective for specific patient groups [41].

This explorative economic evaluation had a number of limitations which should be considered. There was a small sample size in both the robotics cohort (n = 20) and the usual care cohort (n = 10). It is noted that the small sample size was based on the pragmatic implementation of the robotics program, using a sample of convenience (i.e., patients who could be recruited at the time), rather than being based on a pre-determined power or sample size calculation. Missing QOL data for the usual care group meant that a meaningful comparison between groups was not possible. There were also differences at baseline between the two cohorts. Despite the limitations in the data, the findings suggest that it can be feasible to deliver RBT within stroke rehabilitation in a regional public health service. Whilst state-wide scaling of RBT programs may be considered for other regional and metropolitan hospitals, it will be important to undertake demand and feasibility analysis before progressing. Additionally, ongoing collection of outcome data in well-designed clinical trials is critical to advance the current evidence base for robot-assisted therapies in stroke rehabilitation. Current Australian guidelines offer limited guidance for integrating RBT into usual care [42].

## 5. Conclusions

The capabilities approach recognises that rural stroke survivors have a larger capability gap in terms of access to resources to realise their freedoms and outcomes. This research demonstrates the economic feasibility of delivering RBT for rehabilitation in a regional public health service in Victoria, Australia. At a service delivery level, an established out-patient RBT program was demonstrated to be cost-effective when compared to usual care out-patient rehabilitation, specifically requiring less cost for the same effect. At a societal level, by coupling the capabilities approach with economic evaluation methods, the findings from this study can inform public policy, including government investment in rehabilitation services and technologies [23]. This study also offers preliminary evidence linked to global calls to strengthen rehabilitation in healthcare systems [1]. Finally, and importantly, it demonstrates a reduction in the capability gap between rural and metropolitan stroke survivors through the provision of a program that tackled access disadvantage.

**Author Contributions:** Conceptualization, N.B., A.V., M.N., M.M., L.C. and N.L.; methodology, N.B. and N.L; software, N.B. and N.L.; validation, N.B., A.V., M.N., L.C. and N.L.; formal analysis, N.B., A.V. and M.N.; investigation, N.B., A.V., M.N., M.M., L.C. and N.L.; resources, A.V. and M.N.; data curation, N.B., A.V. and M.N.; writing—original draft preparation and review and editing, N.B., A.V., M.N., M.M., L.C. and N.L.; visualization, N.B., A.V., M.N., M.M., L.C. and N.L.; supervision, N.B. and M.N.; project administration, A.V. and M.N.; funding acquisition, A.V. and M.N. All authors have read and agreed to the published version of the manuscript.

**Funding:** This research was funded by a Better Care Victoria Innovation Grant 2016/17.

**Institutional Review Board Statement:** Ethical review and approval were waived for this study, as this was a quality improvement project aiming to embed RBT into usual care.

**Informed Consent Statement:** Patient consent was waived, as this was a quality improvement project aiming to embed RBT into usual care.

**Data Availability Statement:** Due to the waiver of ethical approval, data are not available for this study.

**Conflicts of Interest:** The authors declare no conflict of interest. The funding sponsors had no role in the design of the study; in the collection, analyses, or interpretation of data; in the writing of the manuscript; and in the decision to publish the results.

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
