# Peer review of "Meeting Unmet Needs for Stroke Rehabilitation in Rural Public Health: Explorative Economic Evaluation of Upper Limb Robotics-Based Technologies through a Capabilities Lens"

_societies, doi:10.3390/soc12050143_

Round 1

Reviewer 1 Report

This work is an interesting comparison between robotic therapy and standard care in terms of both economical impact and rehabilitation outcome for people with less possibility to access state-of-the-art technology. In particular, it addresses people living in rural area of Australia. I found the idea interesting and definitely filling a gap in literature. However, neither the results nor the discussion read as smooth and easy to understand as the introduction, which is instead very clear and straightforward. I got lost in the multitude of sub-paragraphs, tables, and I would prefer to have a more concise approach. I believe many of the information presented could be part of supplementary material in order to keep the overall narrative more fluid and simpler, and the message easier to understand. Following, a list of other major and minor comments.

Major

- The sample size, as mentioned by the authors, is a limitation of the study, given its exploratory nature. This is totally fine with me. However, once you start including statistical analysis, I would expect to see Power analysis value together with p-value, and I’m afraid you want have enough power (P>.80) to really state anything, given the small size.

- Given we are comparing 6 robotic programs with standard care, it would be interesting to understand how these programs differ from each other.

Minor

- Not sure if it’s a problem of the specific journal format, but throughout the whole paper many words have unnecessary hyphenation (e.g. Abstract, line 2, “interven-tions”).  

-Tables are in general a bit hard to read. I would personally add some horizontal lines to better separate cells.

- Figure quality is pretty low. All excel plots are stretched and this affects quality. Please consider improving the figures.

Author Response

Comments and Suggestions for Authors

This work is an interesting comparison between robotic therapy and standard care in terms of both economical impact and rehabilitation outcome for people with less possibility to access state-of-the-art technology. In particular, it addresses people living in rural area of Australia. I found the idea interesting and definitely filling a gap in literature. However, neither the results nor the discussion read as smooth and easy to understand as the introduction, which is instead very clear and straightforward. I got lost in the multitude of sub-paragraphs, tables, and I would prefer to have a more concise approach. I believe many of the information presented could be part of supplementary material in order to keep the overall narrative more fluid and simpler, and the message easier to understand. Following, a list of other major and minor comments.

RESPONSE: Thankyou for reviewing this manuscript and providing such valuable feedback. As suggested, we have revised the results whilst ensuring that we offer continued compliance with reporting of the economic evaluation according to the CHEERS checklist for economic evaluations.  Based on reviewer feedback, we have undertaken a major revision of the discussion to improve readability and make the paper more straightforward (including removing some sub-headings).

 Major

- The sample size, as mentioned by the authors, is a limitation of the study, given its exploratory nature. This is totally fine with me. However, once you start including statistical analysis, I would expect to see Power analysis value together with p-value, and I’m afraid you won’t have enough power (P>.80) to really state anything, given the small size.

RESPONSE: You are correct in your observation. This was a pragmatic implementation, based on a sample of convenience (i.e. patients who could be recruited at the time), not based on a pre-determined Power or Sample Size calculation. To ensure transparency for the reader, we have added the word “explorative” after economic evaluation to the manuscript title, consistent with the wording we had already used in the main text. We have also noted this in the limitations.

- Given we are comparing 6 robotic programs with standard care, it would be interesting to understand how these programs differ from each other.

RESPONSE: The following has been added to the results section. “As each iteration of the program passed, there were minor modifications to the program. This includes a change in the staffing profile (from more senior to less senior staff present), an overall reduction in staff present, as well as more patients attending per program. (Figure 2 and 3).”

Minor

- Not sure if it’s a problem of the specific journal format, but throughout the whole paper many words have unnecessary hyphenation (e.g. Abstract, line 2, “interven-tions”).  

RESPONSE: This is beyond our control as it is auto formatted by the template we must use for the journal.

-Tables are in general a bit hard to read. I would personally add some horizontal lines to better separate cells.

RESPONSE: Horizontal lines have been added.

- Figure quality is pretty low. All excel plots are stretched and this affects quality. Please consider improving the figures.

RESPONSE: To improve clarity, Figures 2 to 4 have been added to the end of the manuscript in an editable format.

Reviewer 2 Report

This paper investigates and analyzes the cost-effectiveness of introducing robots in the stroke rehabilitation process. I think this paper has valuable data that demonstrates the superiority of robots, and it is suitable for publication.

I think it is necessary to explain what kinds of robots were used in the rehabilitation process. There are various types and roles of robots. Cost-effectiveness would vary depending on the type of robot.

I did not understand what Figure 2 indicates. I don't think the explanations in the text correspond to the figure.

In the discussion, there is no mention of the fact that the subject of this study is the rehabilitation of the upper limb of stroke patients. I recommend to discuss whether the findings of this study apply to rehabilitation of other functions of stroke patients and rehabilitation of other disease patients, citing examples from previous studies.

Author Response

Comments and Suggestions for Authors

This paper investigates and analyzes the cost-effectiveness of introducing robots in the stroke rehabilitation process. I think this paper has valuable data that demonstrates the superiority of robots, and it is suitable for publication.

RESPONSE: Many thanks for your comments.

I think it is necessary to explain what kinds of robots were used in the rehabilitation process. There are various types and roles of robots. Cost-effectiveness would vary depending on the type of robot.

RESPONSE: This is a good point. We have noted in the methods that the Types of robotics used are detailed in Table 3.

I did not understand what Figure 2 indicates. I don't think the explanations in the text correspond to the figure.

RESPONSE: My apologies – the text was referring to Figure 5, not 2. This has been corrected.

In the discussion, there is no mention of the fact that the subject of this study is the rehabilitation of the upper limb of stroke patients. I recommend discussing whether the findings of this study apply to rehabilitation of other functions of stroke patients and rehabilitation of other disease patients, citing examples from previous studies.

RESPONSE: As suggested, we have undertaken a major revision of the discussion to improve readability and ensure greater comparison with the literature.

Round 2

Reviewer 1 Report

I thank the authors for addressing all the previous concerns I had. I believe the manuscript has improved and it's acceptable in the present form.